# Characterization of Anticancer Effects of the Analogs of DJ4, a Novel Selective Inhibitor of ROCK and MRCK Kinases

**DOI:** 10.3390/ph16081060

**Published:** 2023-07-26

**Authors:** Vijay Pralhad Kale, Jeremy A. Hengst, Arati K. Sharma, Upendarrao Golla, Sinisa Dovat, Shantu G. Amin, Jong K. Yun, Dhimant H. Desai

**Affiliations:** 1Department of Pharmacology Penn State College of Medicine, Hershey, PA 17033, USAjhengst@pennstatehealth.psu.edu (J.A.H.); sga3@psu.edu (S.G.A.); 2Department of Medicine, Penn State College of Medicine, Hershey, PA 17033, USA; ugolla@pennstatehealth.psu.edu; 3Department of Pediatrics, Penn State College of Medicine, Hershey, PA 17033, USA; sdovat@pennstatehealth.psu.edu

**Keywords:** ROCK, MRCK, multi-kinase inhibitor, cancer, invasion, metastasis

## Abstract

The Rho associated coiled-coil containing protein kinase (ROCK1 and ROCK2) and myotonic dystrophy-related Cdc-42 binding kinases (MRCKα and MRCKβ) are critical regulators of cell proliferation and cell plasticity, a process intimately involved in cancer cell migration and invasion. Previously, we reported the discovery of a novel small molecule (DJ4) selective multi-kinase inhibitor of ROCK1/2 and MRCKα/β. Herein, we further characterized the anti-proliferative and apoptotic effects of DJ4 in non-small cell lung cancer and triple-negative breast cancer cells. To further optimize the ROCK/MRCK inhibitory potency of DJ4, we generated a library of 27 analogs. Among the various structural modifications, we identified four additional active analogs with enhanced ROCK/MRCK inhibitory potency. The anti-proliferative and cell cycle inhibitory effects of the active analogs were examined in non-small cell lung cancer, breast cancer, and melanoma cell lines. The anti-proliferative effectiveness of DJ4 and the active analogs was further demonstrated against a wide array of cancer cell types using the NCI-60 human cancer cell line panel. Lastly, these new analogs were tested for anti-migratory effects in highly invasive MDA-MB-231 breast cancer cells. Together, our results demonstrate that selective inhibitors of ROCK1/2 (DJE4, DJ-Allyl) inhibited cell proliferation and induced cell cycle arrest at G2/M but were less effective in cell death induction compared with dual ROCK1/2 and MRCKα/β (DJ4 and DJ110).

## 1. Introduction

Cancer is the second leading cause of death in the USA [1]. Sustained cell proliferation, resistance to cell death, and acquisition of invasive and metastatic phenotypes are some of the fundamental hallmarks of cancer [2]. During the process of metastasis, tumor cells migrate and invade the extracellular matrix [3]. The process of migration is dynamic and requires active remodeling of actin cytoskeleton stress fibers or cortical actin [4,5].

Stress fibers are primarily composed of filamentous actin (F-actin) and myosin II. Stress fibers not only define the shape of the cells but are also responsible for generating contractions of the cell body during migration. Such contractile forces are generated by phosphorylated myosin light chain (MLC) of non-muscle myosin II. MLC phosphorylation is regulated by the Rho associated coiled-coil containing protein kinases (ROCK, Rho kinase; subtypes ROCK1 and ROCK2) and myotonic dystrophy-related Cdc-42 binding kinases (MRCK; subtypes MRCKα and MRCKβ) [6,7,8].

Cancer cell migration is broadly classified into individual (amoeboid and mesenchymal) and collective cell migration. ROCK and MRCK are known to play distinct and compensatory roles in these types of cell migration mechanisms and plasticity of cancer cell migration [9,10]. ROCK also induces contraction of the constriction ring resulting in the successful completion of cytokinesis and division of cells into daughter cells.

Due to the role of ROCK and MRCK in cancer cell proliferation and metastasis, we developed small molecules to simultaneously inhibit both of these targets. Previously, we reported the discovery of DJ4, a novel multi-kinase selective inhibitor of ROCK and MRCK, which is effective against migration and invasion of multiple types of cancer cells [11]. Herein, we examined the anti-proliferation and pro-apoptotic effects of DJ4 in non-small cell lung cancer (NSCLC) and MDA-MB-231 breast cancer cells. We then further optimized DJ4 to develop more potent analogs. In the process of optimization, we discovered four additional active analog compounds. The DJ4-analogs were studied for their in vitro (cell-free) kinase inhibitory activity, anti-proliferative, and pro-apoptotic effects primarily in MDA-MB-231 breast cancer cells. Additionally, all the compounds were investigated for their effects on the cancer cell viability in National Cancer Institute’s (NCI) 60 human cancer cell line screening program. This screening identified the sensitivity of different types of cancer cell lines for these analogs.

Twenty-seven (27) analogs of DJ4 were designed by either substituting or deleting various chemical moieties on group 1 and group 2 (phenethylamine) (Figure 1). Detailed modifications of the functional groups of DJ4 and chemical synthesis are beyond the scope of this manuscript and will be discussed separately. Analogs, including DJ4, were initially tested at 1 µM concentration for their ROCK1 kinase inhibitory activity and four analogs were identified as inhibitors of ROCK1.

Figure 1 depicts general modifications on phenethylamine group of DJ4 that resulted in four active analogs. The substitution of n-ethylbenzene moiety of phenethylamine group (Group 2) with allyl, ethyl, or morpholine moieties resulted in DJ-Allyl, DJE4, and DJ-Morpholine, respectively. DJ110 was designed by reducing one carbon atom from the phenethylamine group of DJ4.

## 2. Results

### 2.1. DJ4 Decreases Survival in Various NSCLC and Breast Cancer Cell Lines

In our previously reported studies, we observed that DJ4 treatment exhibited minor effects on proliferation of A549 and MDA-MB-231 cells after 24 h of exposure [11]. Interestingly, we observed much more profound effects on cell viability in acute myeloid leukemia (AML) cell lines at 24 h [12]. To determine whether the effects on cell viability in AML cells were related to the specific cancer subtypes, or to differences in temporal response to DJ4, we examined whether DJ4 affected the viability of various NSCLC cells over longer periods of drug exposure. MTT assays were performed on H1299, H226, A549, H522, H23, and H460 NSCLC cell lines. After a 72-h treatment, a concentration-dependent decrease in cell survival was observed in all the cell lines. The H1299 cell line was the most sensitive (IC_50_ = 0.44 µM) while the H460 was relatively resistant (IC_50_ = 9.53 µM) (Figure 2A, Table 1). Similarly, the cell viability of MDA-MB-231 breast cancer cells was assessed by counting live/dead cell populations after a 24-h treatment with 1–10 µM concentrations of DJ4. A concentration-dependent increase in cell death from 29% (1 µM) to 47% (10 µM) was observed (Figure 2B). Together, these results indicate that DJ4 decreases the survival of different types of NSCLC and breast cancer cells.

### 2.2. DJ4 Inhibits Colony Forming Ability of Cancer Cells

To further determine the effects of DJ4 on cell proliferation, we performed colony forming assays in A549 lung cancer cells. Forty-eight hours after the treatment with 2.5, 5.0, and 10 µM DJ4, individual cells were re-plated at low seeding density and colony formation was monitored for 14 days. The percent surviving fraction of cells relative to control cells that formed the distinct colonies were 2.3%, 0.4%, and 0.1% at 2.5, 5.0, and 10 µM, respectively, after 14 days of growth in a treatment-free culture medium (Figure 2C,D). Consistent with the observed effect on cell viability, these results indicate that DJ4 significantly inhibits the colony forming ability of A549 lung cancer cells within 48 h.

### 2.3. DJ4 Treatment Arrests Cancer Cells in G2/M Phase

Since DJ4 inhibits phosphorylation of MLC, we hypothesized that DJ4-treated cells are unable to complete cell division and arrest in G2/M phase of the cell cycle. To test this hypothesis, A549 lung cancer cells were synchronized in the G1 phase by serum-starvation for 24 h. After serum replenishment, cells were further treated with 2.5 µM, 5 µM, and 10 µM DJ-4 for 16-, 24-, and 48 h (Figure 3A). Before treatment with DJ4 (time zero), about 80% of cells were in the G1 phase, 20% in the G2/M phase, and essentially none in the S phase. In vehicle treated control cells, there were minimal cells in the G2/M phase at both 24 h (1%) and 48 h (8%). However, DJ4 treatment significantly and dose-dependently increased the G2/M cell population at 2.5 µM (31%), 5 µM (42%), and 10 µM (45%) concentrations at 24 h (Figure 3A). At 48 h, cells remained arrested in the G2/M phase, but also, there was an increase in the sub-G1 cell population indicating apoptotic cells. Together, these results indicate that DJ4 induces a G2/M phase cell cycle arrest consistent with the known roles of ROCK kinases during cytokinesis.

### 2.4. DJ4 Induces Intrinsic Apoptotic Signaling Pathway

G2/M phase arrest commonly precedes induction of the intrinsic apoptotic pathway. We therefore investigated whether DJ4 induced the cell death through activation of this apoptotic pathway (Figure 3). A549 lung cancer cells were treated with 1.25, 2.5, and 5.0 µM concentrations for 16-, 24-, and 48 h. Apoptotic cell death was assessed by fluorescent staining for cell surface exposure of annexin V. 

In response to DJ4 treatment, there was a statistically significant increase in annexin V positive cells. There was a statistically significant increase in annexin V positive cells at 5 µM at 16 h, 2.5 µM and 5 µM at 24 h, and at all the concentrations at the 48-h duration (Figure 3B). Similarly, DJ4 treatment (1.25 µM and 2.5 µM) significantly increased the percentage of annexin V positive cells undergoing apoptosis in MDA-MB-231 breast cancer cells (Figure 3C). Additionally, duplicate samples were studied for mitochondrial membrane depolarization. The depolarization of the mitochondrial inner transmembrane potential is often associated with mitochondrial outer membrane permeabilization (MOMP), release of cytochrome *c* from the mitochondria, and activation of downstream intrinsic apoptotic pathways culminating in activation of executory caspases 3/7 [13,14]. DJ4 (1.25 µM and 2.5 µM) significantly increased the number of cells with depolarized mitochondrial inner membrane indicating activation of the intrinsic apoptosis pathway (Figure 3C). Together, these results suggest that DJ4 induces cell death by activation of the intrinsic apoptotic pathway in NSCLC and breast cancer cells.

### 2.5. DJ4 Modulates Expression of Cell Cycle- and Apoptosis-Regulatory Proteins

To better understand the anti-proliferative and pro-apoptotic effects of DJ4, expression of key proteins involved in proliferation and apoptosis were determined by a Western blot. A549 cells were treated with 2.5 µM and 10 µM concentrations for 24 h and 48 h. Counterintuitively, 2.5 µM of DJ4 significantly induced activation of cell proliferation promoter Erk1/2 (phospho-Erk; pERK) at 24 h, while at 48 h, pERK levels were decreased but were still elevated compared to the vehicle controls (Figure 4A). Interestingly, activated ERK has been shown to induce the expression of the cell cycle inhibitor p21 ^WAF1/CIP1^ (cyclin-dependent kinase inhibitor) correlating with the induction of growth arrest [15]. To this end, expression of p21 ^WAF1/CIP1^ is upregulated at 2.5 µM concentration at 24 h mirroring the activation of pERK (Figure 4A). We also examined the expression of cyclin D3 and CDK4 (G1/S cell cycle checkpoint kinase). Both cyclin D3 and CDK4 were upregulated at 2.5 µM concentration at 24 h which was decreased after a 48-h treatment. The expression of p21 ^WAF1/CIP1^, pERK, CDK4, and cyclin D3 decreased below vehicle treatment levels at both 24 h and 48 h at 10 µM.

To extend these observations to MDA-MB-231 breast cancer cells, cells were treated with DJ4 at 2.5, 5.0, and 10 µM for 24 h. ROCK1/2 catalyzes the inactivating phosphorylation of the myosin phosphatase subunit MYPT1 at Thr696 and Thr853 [16]. To confirm the targeting of ROCK1/2 by DJ4 in MDA-MB-231 cells, we probed for phosphorylation of MYPT1. A concentration-dependent decrease in pMYPT1 (Thr696 and Thr853) was observed, which indicates inactive/less active ROCK in the DJ4-treated cells (Figure 4B). As observed in A549 cells, DJ4 significantly increased the expression of pErk in a concentration-dependent manner in MDA-MB-231 cells (Figure 4B). Cyclin D3 expression was maintained or slightly increased at 2.5 µM and 5 µM as compared to vehicle (DMSO) control and reduced at 10 µM at 24 h.

Among apoptotic markers, DJ4 significantly increased the expression of cleaved caspase 7 at all the concentrations with corresponding increases in caspase substrate, cleaved PARP. This observation indicates that DJ4 induces apoptosis in MDA-MB-231 cells at all the concentrations tested (Figure 4C). In contrast, expression of proliferating cell nuclear antigen (PCNA) protein was decreased at all the concentrations (Figure 4C). PCNA is highly expressed during active DNA synthesis (S phase) [17]. Decreased PCNA expression after DJ4 treatment indicates that the cells are not in active DNA synthesis phase.

Together, these results indicate that DJ4 affects multiple cell cycle checkpoints, however, the observed G2/M phase arrest may predominate over the long-term due to the requirement of actin filaments for cytokinesis.

### 2.6. Assessment of the Kinase Specificity of DJ4

We previously demonstrated that DJ4 is an ATP competitive inhibitor and is thus susceptible to off-target inhibition of other protein kinases [18]. To examine specificity of DJ4 to different human kinases, we performed a KINOMEscan profiling assay (DiscoverX, Fremont, CA, USA) [19]. The KINOMEscan assay consists of a panel of more than 480 kinases of different classes (TK/TKL/STE/CK1/AGC/CAMK/CMGC/others), including clinically relevant mutants, atypical kinases, lipid, and pathogen kinases. A cut-off of %Ctrl score 15 (>85% inhibition) resulted in ~80 kinases, where the majority of them belong to the AGC and CMGC classes, with a primary concentration of 10 µM DJ4 (Figure 5A,B). The potential target kinases, including ROCK1/2, MRCK1/2, FLT3, KIT, and CDK7/8/9/11/14 exhibited more than 98% inhibition in the activity with DJ4 and stood out as hits (Figure 5B). Interestingly, the inhibitory activity of DJ4 is enhanced by certain clinically relevant kinase mutations (Figure 5C,D). For instance, mutations of RET (rearranged during transfection) tyrosine kinase (V804L/V804/M918T) exhibited stronger binding (i.e., higher inhibition) with DJ4 (Figure 5C). Similarly, mutants of the tyrosine kinase epidermal growth factor receptor (EGFR; G718C, G719S, L858R, etc.) exhibited differential inhibition by DJ4 relative to its wild-type form (Figure 5D). Together with our previous studies indicating that DJ4 has potential therapeutic efficacy in mouse models of AML, we determined that targeting the ROCK1/2 and MRCK α/β kinase simultaneously is an effective strategy to reduce migration of highly invasive/metastatic cancers and induce cancer cell death. However, we also recognized the limitations of DJ4 that preclude its clinical development and the need for further optimization of the DJ4 chemotype.

### 2.7. Optimization of ROCK and MRCK Kinase Inhibitory Potential through Development of DJ4 Analogs

With the aim to increase the efficacy and potency of the parent compound DJ4, we generated a library of 27 analogs of DJ4 with substitutions of chemical groups 1 and/or 2 (Figure 1). Cell-free biochemical kinase inhibitory activity of these compounds was evaluated at 1 µM concentration using the previously described ROCK1 kinase activity assay [11]. Among the 27 analogs, we identified four analogs (DJ110, DJ-Allyl, DJE4, and DJ-Morpholine) with >50% inhibition of MYPT1 phosphorylation (Appendix A). These four compounds were selected for further evaluation at 1 µM and 0.1 µM for their in vitro inhibitory activity against ROCK1, ROCK2, MRCKα, and MRCKβ kinases. Qualitative immunoblot analysis indicated that DJ4 and DJ110 significantly inhibit the activity of all the four kinases at 1 µM (Figure 6). DJ110 significantly inhibited activity of the ROCK1, ROCK2, and MRCKα as low as 0.1 µM, whereas DJ4 was ineffective against these kinases at this concentration. The other three analogs, DJE4, DJ-Allyl, and DJ-Morpholine almost completely inhibited the activity of ROCK1 and ROCK2 at 1 µM, similar to DJ4, whereas they displayed minimal inhibition of MRCKα and MRCKβ at this concentration. Together, these results indicated that DJ4 and DJ110 are effective inhibitors of all four kinases whereas DJE4, DJ-Allyl, and DJ-Morpholine are more selective towards ROCK1 and ROCK2.

### 2.8. DJ4 Analogs Effectively Inhibit Cancer Cell Growth

To correlate the observed inhibition of ROCK1/2 and MRCKα/β by the DJ4 analogs on cancer cell growth and determine the relative efficacy of each analog, melanoma (A375M), lung (A549), and breast cancer (MDA-MB-231) cells were treated for 48 h with multiple concentrations of each compound. All compounds, except DJ110 and DJ-Morpholine exhibited a concentration-dependent decrease in cell growth in all the three cell lines (Figure 7). In the MDA-MB-231 cell line, DJ110 significantly decreased the cell growth in a concentration-dependent manner up to 1.25 µM. However, as the concentration increased from 2.5 µM onwards, the effect was reversed. Similarly, DJ-Morpholine decreased the cell growth by ~30% at 0.6215 µM, while the effect remained saturated as the concentration increased. This trend was also observed in A549 lung cancer and A375M melanoma cells. These results suggest that DJ110 and DJ-Morpholine may have limited aqueous solubility. We observed poor solubility of DJ 110 and DJ-Morpholine in 100% DMSO. In contrast, DJ4, DJE4, and DJ–Ally compounds have much better solubility in 100% DMSO. As summarized in Table 2, in spite of its limited solubility, DJ110 was two–five fold more effective than DJ4 or the other analogs.

### 2.9. DJ4 Analogs Induce Cell Cycle Arrest

To confirm the effect of the DJ4 analogs on cell cycle progression, MDA-MB-231 breast cancer cells were treated with each compound at 2.5 µM for 24 h and cell cycle analysis was performed by flow cytometry. All the compounds, except DJ-Morpholine induced significant accumulation of cells in the G2/M phase compared to the vehicle control (Figure 8A). The highest number of cells arrested in the G2/M phase were observed in the DJ110 treatment (42.51%) followed by DJ4 (34%), DJ-Allyl (28.54%), and DJE4 (22.37%) as compared to the control (7.60%). These compounds also exhibited a marginal increase in the sub-G1 apoptotic cell population. However, DJ-Morpholine did not induce cell cycle arrest, possibly due to poor solubility.

### 2.10. DJ4 Analogs Induce Apoptotic Cell Death

To determine whether the active analogs induce apoptotic cell death similar to the DJ4 parent compound, we treated MDA-MB-231 breast cancer cells with 2.5 µM concentration of all the analogs for 48 h (Figure 8). The cells were then analyzed for activation of caspase 3/7 (a marker for apoptotic cell death) and propidium iodide (a marker of cell membrane integrity) using the Muse^TM^ (EMD Millipore) two color bench-top flow cytometer. In Figure 8B, the lower right (Q2) and upper right (Q3) quadrants indicate early apoptotic and late apoptotic cell populations, respectively. Increased apoptosis (Q2 + Q3) was observed in cells treated with DJ110 (64.4%) compared to DJ4 (52.9%), with a shift from early apoptosis toward late apoptosis in response to DJ110. Relative to vehicle treatment, there was also an increase in the upper left quadrant (Q4) cell population observed in response to DJ4 (5.45%) and DJ110 (9.05%) treatment indicative of necrotic cell death. Neither DJ-Allyl (18.1%), DJE4 (13.4%), nor DJ-Morpholine (8%) induced significant apoptotic cell death, further suggesting that targeting ROCK1/2 selectively inhibits cancer cell migration whereas multi-kinase inhibition of ROCK1/2 and MRCK α/β leads to cell death.

### 2.11. NCI-60 Human Cancer Cell Line Screening

To study the effectiveness of these analogs in a vast range of cancer cell lines, the compounds were subjected to cell viability screening in the NCI-60 human cancer cell line panel representing nine different cancer types (Appendix A). Testing of the compounds in such a wide range of cancer types helps identify sensitivity and selectivity of specific types of cancers to the novel compounds. The initial screening was performed at a single concentration of 10 µM. At 10 µM, both DJ110 and DJ-Morpholine were insoluble and were, therefore, ineffective. DJ4 and the other active analogs were effective against multiple cancer types (summarized in Table 3). The parental DJ4 compound was effective in a wide spectrum of cancer cell lines, with breast cancer, NSCLC, and central nervous system cancers displaying the most consistent response. The ovarian cancer cell line NCI/ADR-RES and HCC-2998 colon cancer cells were relatively resistant to DJ4 (IC_50_ > 10 µM) (Appendix A). In general, DJE4 exhibited strong growth inhibitory effects in CNS and breast cancer cell lines. NCI/ADR-RES (ovarian), RPMI-8226, and K-562 (leukemia) cell lines were relatively resistant (29%, 29%, 26% increase in cell number) to DJE4. DJ-Allyl was the most effective in CNS and breast cancer cell lines. Two CNS cell lines, SNB-75 and SF-539, were the most sensitive in which DJ-Allyl induced the cell death (60% reduction in cell number). NCI/ADR-RES (ovarian), MALME-3M (melanoma), HCC-2998 (colon), RPMI-8226, and K-562 (leukemia) were relatively resistant (~40% increase in cell number) to DJ-Allyl. Together, these screening results demonstrate that multiple cancer cell types are sensitive to DJ4 and the tested analogs.

### 2.12. DJ4 Analogs Potently Inhibit Migration of Cancer Cells

Lastly, we examined the effects of these four analogs on migration of MDA-MB-231 breast cancer cells. Scratch assays were performed on cells treated with the various analogs for 24 h and then allowed to migrate into the scratched area for 9 h in compound-free media. Treatment with 2.5 µM of these compounds for 24 h significantly inhibited the migration of cancer cells (Figure 9A). The greatest inhibitory effect on migration was observed with DJ110 (28% migration compared to control) followed by DJ4 (32%), DJ-Allyl (36%), and DJE4 (48%) treatment (Figure 9B). Due to limited solubility, DJ-Morpholine was not tested for its effect on cell migration.

Because DJ4 inhibits cell proliferation, it could be assumed that the anti-proliferative effect of these analogs contributes to the observed inhibition of migration. However, we designed this study to mitigate this possibility by employing a 9-h drug-free exposure period. We reasoned that this short period of migration would be insufficient for MDA-MB-231 cells to significantly proliferate and fill the scratched area as demonstrated in Figure 9A vehicle treatment conditions. Therefore, the anti-migratory effect is likely independent of the anti-proliferative effect of the analogs. To confirm that the anti-migratory effect is independent of the cell death effect, we measured the viability of cells at the end of the duration of migration (9 h). Cells from the scratch assay were stained with calcein (live cells) and propidium iodide (PI, dead cells). In the presence of each DJ4 analog, the majority of the cells were alive (green) in the vicinity of the migration area (Figure 9A). This indicates that the anti-migration effect of the compounds is most likely to be independent of cell death. Together, these studies suggest that DJ4 analogs effectively inhibit migration of cancer cells independent of their growth inhibition and cell death induction effects.

## 3. Discussion

Metastatic cancer cells exhibit a high degree of plasticity in their migratory phenotypes modulated by multiple proteins such as MMP, ROCK, and MRCK [10,20]. Hence, rather than targeting any single protein, inhibition of both ROCK and MRCK might be a more efficient strategy for inhibition of invasion/metastasis [21]. We previously reported that DJ4 targets both ROCK and MRCK kinases, the key proteins responsible for plasticity of cancer cell migration [11]. In this study, we evaluated the cell death induction and anti-cell proliferation effects of DJ4. We further designed various structural analogs of DJ4 to explore the possibility of discovering more potent inhibitors. The process of optimization yielded four additional active compounds. Any modification on chemical group 1 (represented by the green dotted rectangle, Figure 1) resulted in a complete loss of DJ4 activity (unpublished data) which indicates that chemical group 1 is critical for the activity of DJ4. However, the modifications on chemical group 2 (phenethylamine; represented by the red dotted rectangle, Figure 1) resulted in the altered potency and/or target specificity of the analogs. Amongst all the modifications in chemical group 2 of DJ4, deletion of a carbon atom resulted in a more potent compound, DJ110. However, this modification also decreased the solubility, which limits further development of DJ110. Other modifications wherein chemical group 2 was substituted with either an ethyl (DJE4) or morpholine moiety (DJ-Morpholine) were relatively selective towards ROCK1 and ROCK2 only. Substitution with an allyl moiety resulted in a strong inhibitor (DJ-Allyl) of ROCK1, ROCK2, and a relatively weak inhibitor of MRCKα. These analogs were also less potent than both DJ4 and DJ110. DJ-Morpholine significantly inhibited ROCK1/2 kinase activity in a cell-free system. However, it was less effective in all the cellular assays. This could have resulted from the poor solubility of DJ-Morpholine. This finding indicates that morpholine modification decreases the solubility and, hence, cellular effects of DJ4. Additionally, other substitutions (allyl and ethyl) decreased their activity against MRCKα/β. Further studies are warranted to identify docking orientation(s) and understand the structural basis of the preferential selectivity of these analogs for ROCK1/2 and MRCKα/β.

In our earlier report [11], time-lapse microscopy studies indicated that the parent DJ4 compound effectively inhibits proliferation of breast and lung cancer cells. Herein, we further analyzed whether DJ4 and four of its analogs induce cell cycle arrest. The cell cycle analysis indicates that all the DJ4 analogs, except DJ-Morpholine, induce significant arrest of the cell cycle in the G2/M phase. Cell cycle arrest in the G2/M phase may be because of the critical role ROCK plays during cell division. The phosphorylation of MLC of myosin II is required for completion of cell division [10]. ROCK phosphorylates MLC (Ser19) at the contractile ring of the cleavage furrow during cytokinesis [9,22,23]. ROCK also phosphorylates intermediate filaments such as vimentin (Ser71) and glial fibrillary acidic protein (Ser38) during cleavage furrow ingression, which underlies the importance of ROCK during cytokinesis [24,25]. Indeed, silencing of ROCK in *Drosophila* S2 cells impairs cytokinesis during cell division [26]. Hence, dysfunction of either MLC or ROCK impairs cell division and arrests the cells in the G2/M phase of the cell cycle. Treatment of HeLa cells with another ROCK inhibitor, Y27632 (>100 µM), impaired cytokinesis [27]. Additionally, siRNA-mediated silencing of ROCK in HeLa cells induced cell cycle arrest in the G2/M phase [28]. On the contrary, treatment of astrocytes at 10 µM Y27632 promoted proliferation and progression of the cell cycle [29]. Such contradictory observations may be due to the use of low concentrations of ROCK inhibitors, which may not be sufficient to block cytokinesis and induce cell death. In fact, it has been observed that the use of lower concentrations of ROCK inhibitors such as Y27632 delays G1-S transition rather than completely blocking the cells at the G1 phase, while high concentrations effectively inhibit cytokinesis [27]. Additionally, different cell types may have different requirements of ROCK during the progression of the cell cycle. The decreased expression of PCNA supports cell cycle analysis data that DJ4-treated cells are not undergoing active DNA synthesis. Actively proliferating cells have the highest PCNA during the S-phase, while low levels of expression can be observed during other phases of the cell cycle including G2/M [17]. Our studies further suggest that a 48-h treatment of A549 cells with DJ4 significantly reduced the colony forming ability of the treated cells, which is one of the key characteristics of solid tumor cells, indicating that most of the cells do not overcome the cell cycle arrest even after 14 days.

A proposed model for the possible mechanisms of DJ4 mediated modulation of anti-proliferative and pro-apoptotic proteins is depicted in Figure 10.

DJ4 treatment increased the expression of pro-proliferative proteins such as p-Erk, CDK4, and cyclin D3. However, at the same concentration and treatment duration, there is upregulation of CDK inhibitor proteins such as p21 ^WAF1/CIP1^ and activation of caspase 7. This may be because ROCK1 directly interacts and suppresses the activation of Erk1/2 [29]. Indeed, inactivation of ROCK by Y27632 or Rho signaling by selenite hyper-phosphorylates Erk1/2, independent of MEK1/2 [30]. Hyperphosphorylated Erk1/2 transcriptionally activates p21^WAF1/CIP1^ [31]. RhoA inactivation has shown to increase hyperphosphorylation of Erk1/2 and activate p21^WAF1/CIP1^ in Swiss3T3 and NIH3T3 cells [32]. Activated Erk1/2 can directly affect mitochondrial function by decreasing mitochondrial membrane potential [33,34,35]. This could compromise mitochondrial membrane integrity and lead to the release of cytochrome *c* into the cytoplasm [35,36,37]. Cytochrome *c* release is often followed by the activation of the intrinsic apoptosis pathway, which is observed in DJ4 treatment. Hyperphosphorylated Erk1/2 has also been observed to be associated with upregulation of pro-apoptotic Bcl2 family members such as Bax that can induce the release of cytochrome *c* into the cytoplasm [38,39,40]. There is also a possibility that DJ4-treated cells undergo apoptosis due to the combined effect of G2/M arrest, and concurrent increase in both pro-proliferative (pErk, cyclin D, and CDK4) and anti-proliferative proteins (p21^WAF1/CIP1^ and caspase 3/7).

As determined by kinase selectivity profiling, DJ4 is a multi-kinase inhibitor that selectively inhibits AGC and CMGC class kinases, and its activity is altered with clinically significant mutants of some of the kinases. (Figure 4). It is, therefore, likely that the analogs of DJ4 display similar multi-kinase inhibition profiles. Thus, the contribution of inhibition of these other kinases must be considered when examining the mechanism-of-action of DJ4. Altogether, the studies presented previously and, herein, suggest that DJ4 and its analogs are poly-pharmacological agents. The poly-pharmacological nature of DJ4 could contribute to “off-target” toxicity that affects its maximal tolerated dosage. However, like many other “targeted” therapeutic agents, the anti-cancer efficacy of DJ4 may be enhanced by its ability to simultaneously target numerous signaling pathways relevant to cancer cell growth/survival/metastasis. Further efforts are underway to evaluate the DJ4 specificity and develop structure-activity relationship (SAR) models to generate additional analogs of DJ4 with enhanced selectivity, stability, and solubility. The development of more selective DJ4 analogs, coupled with detailed studies employing molecular inhibition strategies, such as Crispr/Cas9 targeting of relevant DJ4 inhibitory targets, will help to further elucidate the roles of ROCK/MRCK in the molecular mechanism of DJ4 analogs.

Using the NCI-60 human cancer cell panel, we explored the sensitivity of DJ4 analogs across a variety of cancer cell lines representing different histology and genotypes. DJ110 and DJ-Morpholine are sparingly soluble in the culture media at 10 µM which makes them ineffective at this concentration. Single concentration (10 µM) screening data suggested DJE4 as more efficacious across the cell lines, however, our detailed multiple studies strongly suggest that DJ4 and DJ110 are the most potent and efficacious compounds. Discordance in the efficacy data between the NCI screening and other in-house studies may be due to differences in methodology and the use of single concentration in NCI screening. The NCI studies used SRB assays to study cell viability, while in-house efficacy studies were performed with MTT, caspase 3/7 activity, cell migration, and cell cycle assays. Secondly, NCI screening was conducted at a single concentration of 10 µM which does not accurately compare the relative potencies of DJE4 and DJ4. The enhanced effectiveness of DJE4 at 10 µM could be attributed to its increased solubility due to the replacement of the hydrophobic phenyl moiety on DJ4 with a hydrophilic ethyl moiety, which can be further supported by the fact that DJE4 is proven to be less efficacious than DJ4 at <5 µM concentrations in cell migration and cell death assays. This solubility effect is more pronounced at higher concentrations where phenyl containing compounds, such as DJ4 and DJ110, have limited solubility. Hence, in this case, NCI’s single dose cell viability data should be interpreted to understand the relative sensitivity of a particular cell line to a compound rather than comparing the efficacy between the compounds.

Cell lines such as HCT-15 (colon cancer), NCI/ADR-RES (ovarian cancer), and CAK-1 (renal cancer) were resistant to DJ4 analogs. These cell lines exhibit high expression of the gene ABCB1 which encodes the multi-drug resistant protein, P-glycoprotein, (P-gp; data retrieved from http://discover.nci.nih.gov/cellminer/analysis.do (accessed on 14 July 2023)) [41,42,43]. Our preliminary studies with vincristine-resistant HL60-VCR leukemic cells confirm that DJ4 analogs are the substrates of P-gp (Appendix A). P-gp pumps the compounds out of the cells and makes it unavailable for cellular action. However, it is difficult to reconcile the relative sensitivity or resistance of other cell lines to DJ4 or its analogs from the current studies. Detailed studies that consider the genetic alterations of individual cell lines are required to explain why cell lines of the same cancer subtype respond to DJ4 and its analogs differently.

## 4. Materials and Methods

### 4.1. Cell-Free (Biochemical) Kinase Activity Assays

Cell-free biochemical kinase activity was assayed using published procedures [11]. Briefly, recombinant ROCK1 (9.48 nM) or ROCK2 (8.26 nM; Invitrogen, Waltham, MA, USA) was incubated in the presence of 0.1 and 1 µM concentrations of DJ4 analogs or DMSO in ROCK-assay buffer at room temperature (RT) for 10 min. MRCKα and MRCKβ (2 ng/µL; Invitrogen) assays were performed in the assay buffer recommended by the manufacturer. Recombinant MYPT1 (20 ng/µL; Millipore) and ATP (5 µM) were added to initiate the reaction. The reaction was incubated at 30 °C for 20 min. Western blot analysis of these samples for phosphorylation of MYPT1 was performed as per the method mentioned in our earlier publication [11]. Briefly, an equal amount of sample was loaded on Bis-Tris gel (Novex, Life Technologies, Carlsbad, CA, USA), proteins were electrophoresed and transferred to PVDF membranes. The blots were probed with anti-pMYPT1 (Thr696) or anti-MYPT1 (Millipore) antibodies.

### 4.2. Cell Lines and Cell Culture

The following cell lines used in this study were obtained from ATCC: NSCLC (A549, CCL-185; H522, CRL-5810; H23, CRL-5800; H2126, CCL-256; H460, HTB-177), melanoma (A375M, CRL-1619), and breast cancer (MDA-MB-231, HTB-26). Cells were maintained in DMEM or RPMI media (Cellgro, Corning, NY, USA) supplemented with 10% fetal bovine serum (Gibco, Fisher Scientific, Hampton, NH, USA) and penicillin/streptomycin (Gibco, Fisher Scientific, Hampton, NH, USA) at 37 °C with 5% CO_2_ humidified incubator. Acute myeloid leukemia cells (AML, HL60; CCL240) were obtained from ATCC while vincristine resistant HL60 (HL60VCR) cells were a gift from Dr. Hong-Gang Wang (Penn State College of Medicine, Hershey, PA, USA) and both the cells were cultured in IMDM medium (HyClone, Washington, D.C., USA) with 10% FBS.

### 4.3. Cell Cycle Analysis

To understand the effect of DJ4 on cell proliferation, cell cycle analysis was conducted in A549 cells. The cells were treated with indicated concentrations for 48 h. For analogs, MDA-MB-231 cells were treated with 2.5 µM of analogs for 24 h. At the end of treatment, media were collected and centrifuged to collect the dislodged cells. The adhering cells were washed with warm Dulbecco’s phosphate buffered saline (DPBS; without calcium) and trypsinized with 0.05% trypsin for 2–3 min followed by gentle scraping. The cells were centrifuged at 200× *g* for 5 min and resuspended in warm 0.5 mL DPBS. During resuspension, the cells were gently pipetted up and down to prepare individual cell suspension. The cells were fixed with 4.5 mL 70% cold ethanol with continuous gentle vortexing. After fixation, the samples were centrifuged to remove methanol and resuspended in DPBS, re-centrifuged and stained with propidium iodide staining solution for 15 min at 37 °C. The cells were filtered to avoid clumps by using round-bottom tubes with cell-strainer cap (Falcon, Fisher Scientific, Hampton, NH, USA). The samples were analyzed for cell cycle using BD FACS Calibur^TM^ (BD Biosciences, Franlin Lakes, NJ, USA) flow cytometer and the data were quantified by ModFit LT ^TM^ V3.3.11 software (Variety Software House, Bedford, MA, USA).

### 4.4. Caspase 3/7 Assay

To examine whether the selected compounds induce apoptosis, we treated MDA-MB-231 breast cancer cells with 5 µM of each compound. After 48 h treatment the apoptotic cells were detected using caspase 3/7 assay kit and analyzed by laser-based fluorescent detection using Muse^TM^ cell analyzer (Millipore Inc., Burlington, MA, USA) as per the manufacturer’s instructions. The data were quantified by using FlowJo software (version 10).

### 4.5. Live/Dead Cell Count

MDA-MB-231 cells were treated for 24 h with 1 µM, 2.5 µM, 5 µM, and 10 µM concentrations of DJ4. Cells were trypsinized with 0.05% trypsin. Floating cells were collected and included in the analysis. Cell number and cell viability were analyzed using Muse^TM^ Cell Viability reagent and Muse^TM^ (Millipore, Inc. Burlington, MA, USA) cell analyzer as per the manufacturer’s instructions.

### 4.6. Annexin V Analysis

To study induction of apoptosis over a period of time, A549 cells were treated with 1.25 µM, 2.5 µM, and 5 µM concentrations of DJ4 for 16-, 24-, and 48 h. MDA-MB-231 cells were treated with 2.5 µM and 5 µM concentrations of DJ4 for 48 h. The cells were trypsinized with 0.05% trypsin. Floating cells were also collected for analysis. The samples were incubated with Muse^TM^ Annexin V reagent and analyzed by laser-based fluorescent detection using Muse^TM^ (Millipore Inc.) cell analyzer per manufacturer’s instructions.

### 4.7. Mitochondrial Membrane Depolarization

Cells undergoing mitochondrial inner membrane depolarization were analyzed to understand the underlying apoptosis pathway of cellular death. MDA-MB-231 cells were treated with indicated concentrations of DJ4 and incubated for 48 h. The cells were trypsinized with 0.05% trypsin. Floating cells were collected for analysis. The samples were incubated with Muse^TM^ Mito Reagents and analyzed by laser-based fluorescent detection using Muse^TM^ (Millipore Inc., Burlington, MA, USA) per manufacturer’s instructions.

### 4.8. Western Blot Analysis

For protein expression studies, MDA-MB-231 cells were treated with indicated concentrations of DJ4 for 24 h. Additionally, A549 cells were treated for 24 h and 48 h. Cell lysate preparation, protein estimation, and Western blot analysis was performed as per the procedure reported earlier [11]. The blots were probed with cleaved caspase 7 (#9492, Cell Signaling, Danvers, MA, USA), GAPDH (#97166, Cell Signaling, Danvers, MA, USA), β-Actin (#3700, Cell Signaling, Danvers, MA, USA), pErk (#5726, Cell Signaling, Danvers, MA, USA), p21 ^WAF1/CIP1^ (#60480, Cell Signaling, Danvers, MA, USA), cyclinD3 (#2936, Cell Signaling, Danvers, MA, USA), CDK4 (#23972, Cell Signaling, Danvers, MA, USA), cleaved PARP (#32563, Cell Signaling, Danvers, MA, USA), pan Erk (#23887, Cell Signaling, Danvers, MA, USA), pMYPT1 (Thr853, # SAB4503944, Millipore, Inc., Burlington, MA, USA) and PCNA (#sc-56, Santa Cruz Biotechnology, Inc., Dallas, TX, USA) antibodies.

### 4.9. Cell Migration Assay

Migration (scratch/wound healing) assays were performed using published procedures [11]. Briefly, MDA-MB-231 cells were grown to confluence, and cells were treated with DMSO (vehicle control), DJ4, and its active analogs at 2.5 µM concentration for 24 h. At the end of the treatment period, uniform scratches were made in the monolayer of cells. The cells were allowed to migrate for 9 h in a compound-free medium. Light microscopic images were obtained at 0 h and 9 h after creating scratches. The width of the scratches was measured by using AxioVision software (AxioVision Inc., Carl Zeiss, Jena, Germany). The percentage of migration in each treatment at 9 h was calculated in comparison to 0 h readings. Furthermore, the percent migration in treatment groups was normalized to vehicle control (considering 100% migration). The experiment was performed in triplicate. Viability of the cells at the migration area was studied to confirm that the anti-migration effect of the analogs at the tested concentration and duration is independent of cell death. At the end of the migration period, cells were stained with calcein (Invitrogen, Waltham, MA, USA) and propidium iodide (Invitrogen, Waltham, MA, USA) dyes, as per the manufacturer’s protocol, which stain the live and dead cells, respectively.

### 4.10. MTT Assay

The cytotoxic effect of DJ4 was studied in H1299, H226, A549, H522, H23, and H460 non-small cell lung cancer cells. The cells were seeded into 96-well plates with confluence (15 × 10^4^ cells/well). The cells were further treated with DMSO or DJ4 for 72 h. To study the effects of analogs, A549, A375M, and MDA-MB-231 cells were plated with low density (2000 cells/well) in a 96-well plate and grown overnight at 37 °C (with 5% CO_2_). Cells were treated with the analogs at the concentration of 0–10 µM for 48 h. At the end of the treatment period, cells were incubated with a tetrazolium dye MTT (3-(4,5-dimethylthiazol-2-yl)-2,5-diphenyltetrazolium bromide) reagent (Sigma) at 37 °C for 3 h to develop formazan crystals. The media was siphoned off and crystals were dissolved in DMSO and mixed gently on rocker for 5 min. Absorbance was read at 570 nm and 630 nm (background) wavelength. IC_50_ values were calculated from non-linear regression curve fit by using GraphPad Prism 5.0 software.

### 4.11. Kinase Inhibition Assay

The KINOMEscan^®^, in vitro competition binding assay, was performed by Lead Hunter Discovery Services (DiscoverX Corporation, Fremont, CA, USA) to determine the kinase specificity of DJ4, a novel kinase inhibitor [19,44]. The KINOMEscan^®^ panel consists of more than 480 kinases, including clinically relevant mutants, lipid, atypical, and pathogen kinases, plus a growing panel of activation-state specific assays. The primary screen was performed using a single concentration of 10 µM, and the results for DJ4 binding interactions are reported as percent of control (%Ctrl). The %Ctrl is calculated as follows: [(DJ4 signal-positive control signal)/(negative control DMSO signal-positive control signal)] x 00. The human kinome tree annotation for DJ4 is plotted with Reaction Biology’s Kinase Mapper and KinMap web interface tool http://kinhub.org/kinmap (accessed on 09 January 2023) [45].

### 4.12. NCI-60 Human Cancer Cell Line Screen

The selected active compounds were screened at NCI, as a part of In Vitro Cell Line Screening Project (IVCLSP), for their effect on cell growth in 60 different cancer cell lines. The 60 human cancer cell line panel included leukemia, melanoma, non-small cell lung cancer, colon, central nervous system (CNS), ovarian, renal, prostate, and breast cancer cell lines. The detailed procedure and applications of NCI-60 cell line screening can be found at https://dtp.cancer.gov/discovery_development/nci-60/default.htm (accessed on 9 January 2023) [46,47,48,49]. Briefly, cells were plated in 96-well plates and incubated for 24 h. One set of cells was fixed with trichloroacetic acid at time 0 h (T_z_), and SRB (sulphorhodamine B) assays were performed to quantify the cells. The remaining set of cells were treated with 10 µM single concentration of each compound and incubated for 48 h (T_e_) after which the cells were fixed, and the SRB assays were performed. Data are presented as percent growth relative to the control as 100% according to the following calculations:

When T_e_ >/= T_z_.
$$Growth(\%)=\frac{T_{e}-T_{z}}{C-T_{z}}*100$$
and when T_e_ < T_z_.
(1)$$Growth(\%)=\frac{T_{e}-T_{z}}{T_{z}}*100$$

Values between 0–100 indicate cell growth, zero (0) indicates no cell growth, values < 0 indicate cell death.

### 4.13. Statistical Analysis

The data were analyzed for statistical significance by one-way or two-way ANOVA and Dunnett’s multiple comparison posttest or Bonferroni posttest using Prism version 5.1 (GraphPad, Inc., Boston, MA, USA). Statistical significance was analyzed at 5% confidence level unless otherwise indicated. IC_50_ values in MTT cell growth assay were determined from a non-linear regression curve using Prism version 5.1.

## 5. Conclusions

In conclusion, our present studies suggest that multi-kinase inhibitors of ROCK and MRCK (DJ4 and DJ110) effectively block cancer cell migration and cell survival better than selective ROCK inhibitors (DJE4, DJ-Allyl, and DJ-Morpholine). Reduction of a single carbon atom from the phenethylamine moiety of DJ4 yielded a novel compound, DJ110, although it also altered the solubility of the compound. Thus, these studies continue to indicate the therapeutic potential of targeting the ROCK and MRCK kinases simultaneously as a generalized anti-cancer strategy that has the added benefit of inhibiting invasion/metastasis. However, additional optimization of the DJ4 chemotype is necessary for the advancement of this therapeutic strategy to clinical application.

## Figures and Tables

**Figure 1 pharmaceuticals-16-01060-f001:**
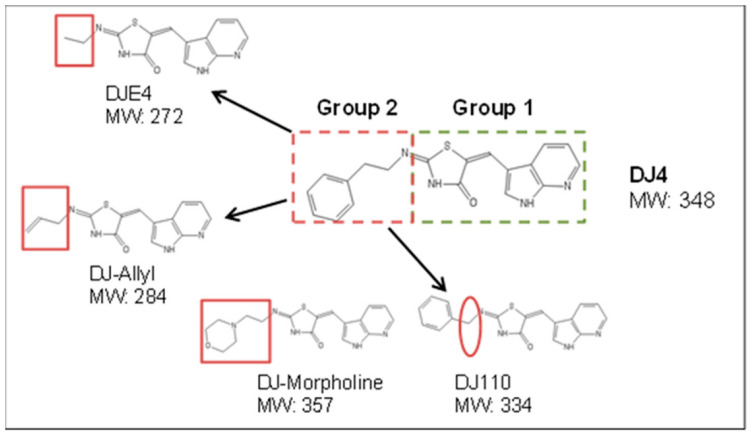
**Structural modifications on phenethylamine group (chemical group 2) of DJ4.** Various chemical groups on DJ4 were modified. However, modifications on chemical group 2 (phenethylamine) resulted in four active molecules. Those modifications are shown in the figure.

**Figure 2 pharmaceuticals-16-01060-f002:**
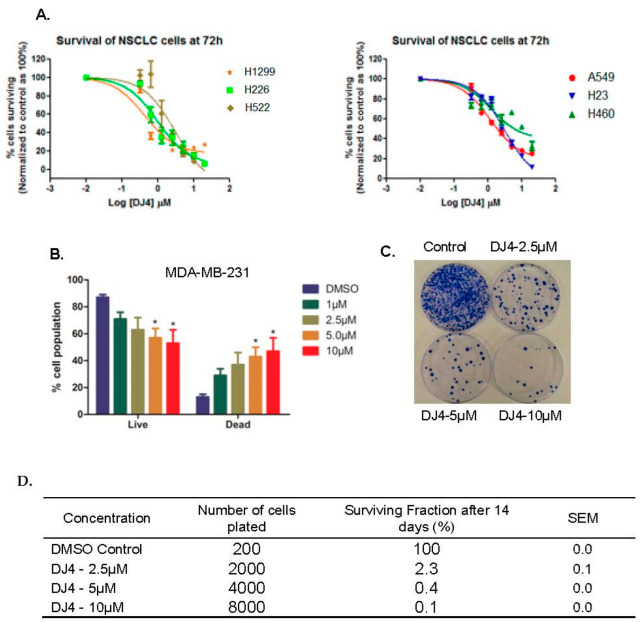
**DJ4 reduces cell survival and inhibits colony forming ability of the cancer cells.** (**A**) Different NSCLC cells were plated in a 96-well plate to confluence and then treated for 72 h. Cytotoxicity was assessed by MTT assay. The error bars indicate SEM of triplicates. (**B**) To assess cytotoxicity of DJ4 in breast cancer cells, MDA-MB-231 cells were treated for 24 h with indicated concentrations of DJ4. Percentage of live cells were counted using Muse^TM^ Live/Dead reagent and analyzed by Muse^TM^ cell analyzer. Error bars indicate SEM. *n* = 2. * *p* < 0.05. (**C**) DJ4 inhibits colony forming ability of A549 cells treated for 48 h. After a 48-h treatment, cells were seeded with number of cells mentioned in the table and were allowed to grow and form colonies. At the end of 14 days, colonies were stained and counted. Percent surviving fraction was calculated. The figure is the representative of the replicates. (**D**) The data are summarized in the table and represent the average of three replicates.

**Figure 3 pharmaceuticals-16-01060-f003:**
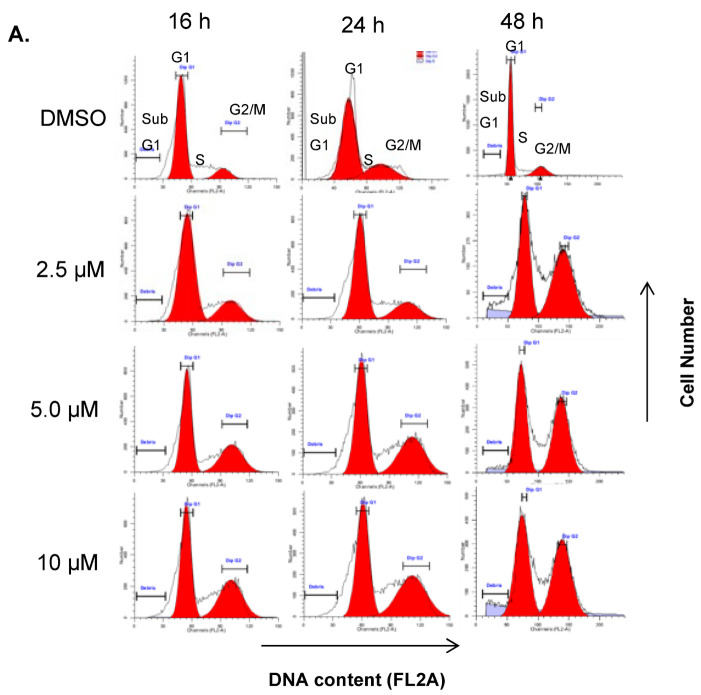
**DJ4 arrests A549 lung cancer cells in G2/M phase and induces apoptotic cell death in NSCLC and breast cancer cells.** (**A**). A549 cells were serum-starved for 24 h and treated for 16 h, 24 h, and 48 h with indicated concentrations. Propidium iodide stained cells were analyzed for DNA content by flow cytometry. The cell cycle graphs indicate distribution of cells in the G1, S, and G2/M phases of cell cycle. Sub-G1 cell population indicates apoptotic cells. (**B**). Annexin V staining of A549 cells. The cells were treated with indicated concentrations and durations. Percent annexin V + cells were analyzed using Muse^TM^ cell analyzer. Error bars indicate SEM. *n* = 2. (**C**). Analysis of annexin V + ve and mitochondrial membrane depolarized cells. MDA-MB-231 cells were treated for 48 h at the indicated concentrations. Staurosporin (2.5 µM) was used as a positive control. Error bars indicate SEM. *n* = 2. * *p* < 0.05.

**Figure 4 pharmaceuticals-16-01060-f004:**
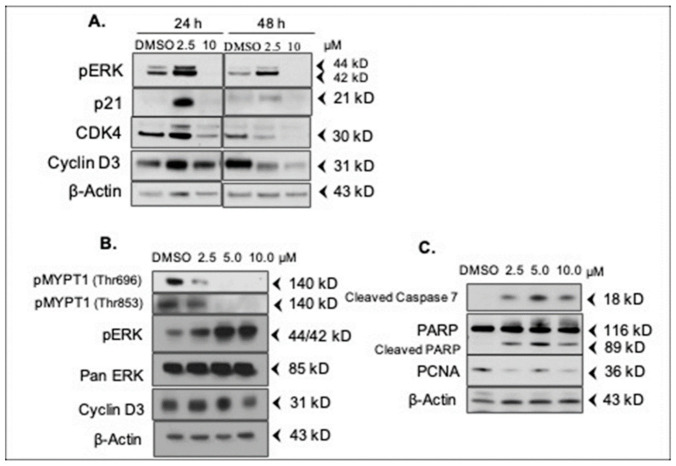
**DJ4 modulates expression of cell cycle- and apoptosis-regulating proteins.** (**A**) A549 lung cancer cells were treated for 24 h and 48 h at 2.5 µM and 10 µM concentrations of DJ4. (**B**,**C**) MDA-MB-231 breast cancer cells were treated for 24 h at the indicated concentrations. Various proteins were immunoblotted with respective antibodies. β-Actin or GAPDH were used as internal loading controls and were probed on each blot. However, here, only one representative blot of β-Actin per panel is shown.

**Figure 5 pharmaceuticals-16-01060-f005:**
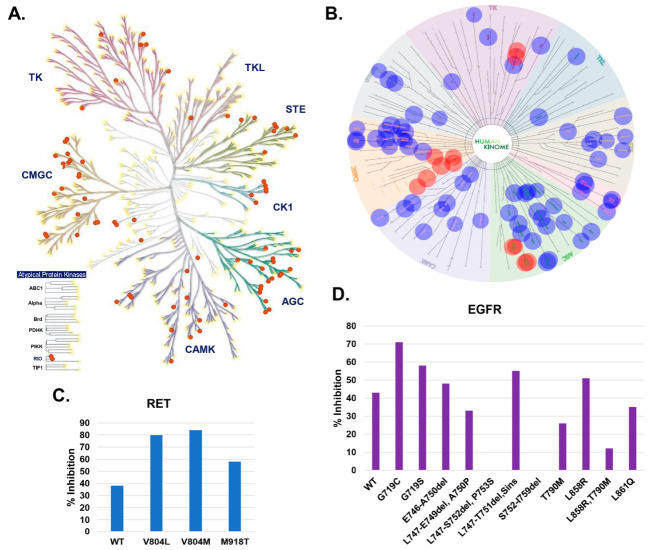
**The kinase selectivity profile for DJ4 assessed using KINOMEscan high-throughput assay.** (**A**) The kinome tree annotation by KinMap for ~80 kinases (red colored dots) that exhibited %Ctrl score over 15 (>85% inhibition) with DJ4. (**B**) The kinases with %Ctrl score greater than 15 (i.e., >85% inhibition with DJ4) were visualized using Reaction Biology’s kinase mapper tool (www.reactionbiology.com/tools/kinase-mapper; accessed on 9 January 2023). The red spots (ROCK/MRCK/CDK/FLT3/KIT) indicate kinases with high specificity to DJ4. (**C**,**D**) The % inhibition profile of DJ4 with wild-type (WT) and its clinically relevant kinase mutants of RET (rearranged during transfection) tyrosine kinase (**C**) and EGFR (epidermal growth factor receptor) protein kinase (**D**).

**Figure 6 pharmaceuticals-16-01060-f006:**
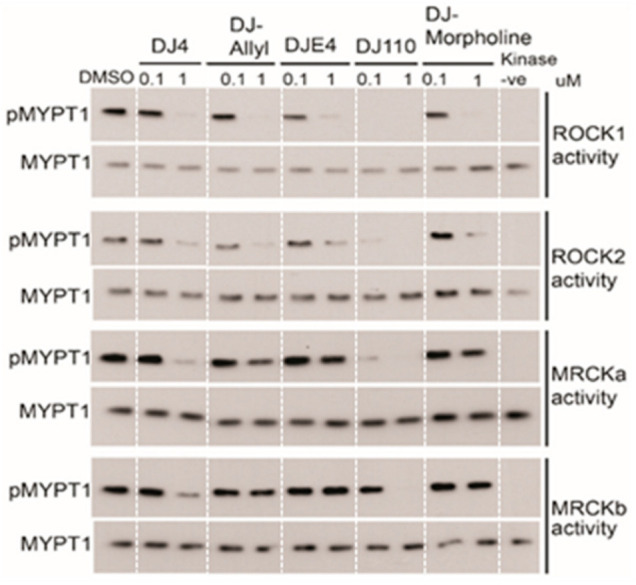
**Chemical optimization of DJ4 and kinase inhibitory activity of active analogs.** DJ4 analogs inhibit kinase activity of either ROCK 1/2 or both ROCK 1/2 and MRCK α/β in cell-free biochemical kinase assay. DJ 4 and DJ 110 inhibited both ROCK 1/2 and MRCK α/β kinases.

**Figure 7 pharmaceuticals-16-01060-f007:**
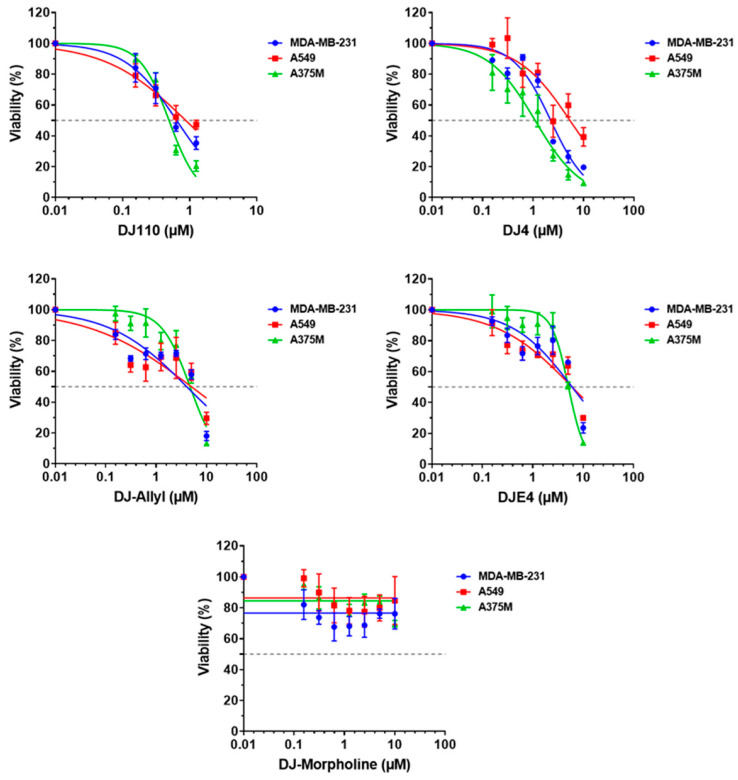
**DJ4 analogs reduce cell growth in melanoma, breast, and lung cancer cells.** MTT assay was performed to evaluate the effect of DJ4 analogs on cell growth after 48 h of treatment. IC50 values were calculated with nonlinear least squares regression curve fit by GraphPad prism. DJ110 and DJ4 were the most potent with the least IC50 concentrations. DJ-Morpholine was the least effective among all. The error bars indicate SD from triplicates of the experiment.

**Figure 8 pharmaceuticals-16-01060-f008:**
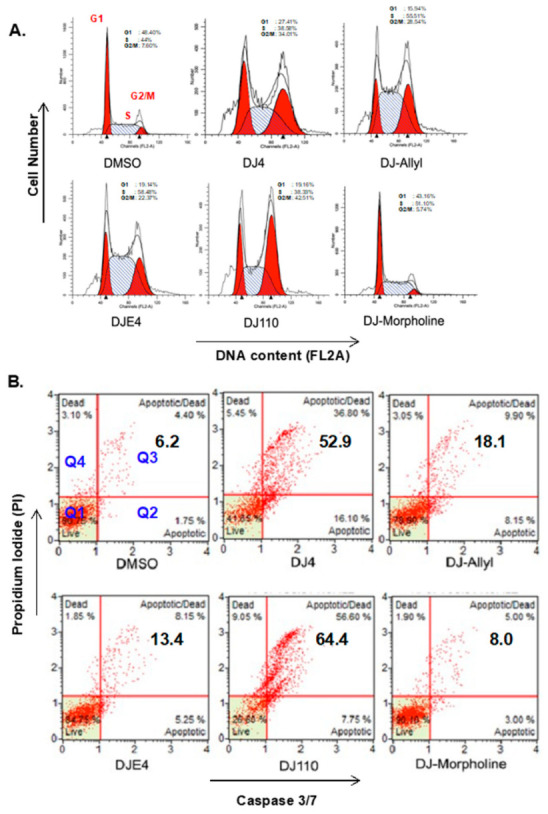
**DJ4 analogs induce cell cycle arrest and apoptotic cell death in MDA-MB-231 breast cancer cells**. (**A**) DJ4 analogs induce cell cycle arrest in MDA-MB-231 breast cancer cells. To study the cell cycle arrest by DJ4 analogs, serum-starved cell cycle-synchronized MDA-MB-231 cells were treated with 2.5 µM concentration of each compound for 24 h and stained with propidium iodide. Treatment with all the analogs, except DJ-Morpholine, induced significant arrest of cells in G2/M phase of cell cycle. (**B**) Apoptotic cell death by DJ4 analogs was evaluated by measuring caspase 3/7 activation in breast cancer cells. Cells were treated with 2.5 µM concentration of each compound for 48 h and incubated with fluorescent dye. All the compounds except DJ-Morpholine induced significant apoptotic cell death.

**Figure 9 pharmaceuticals-16-01060-f009:**
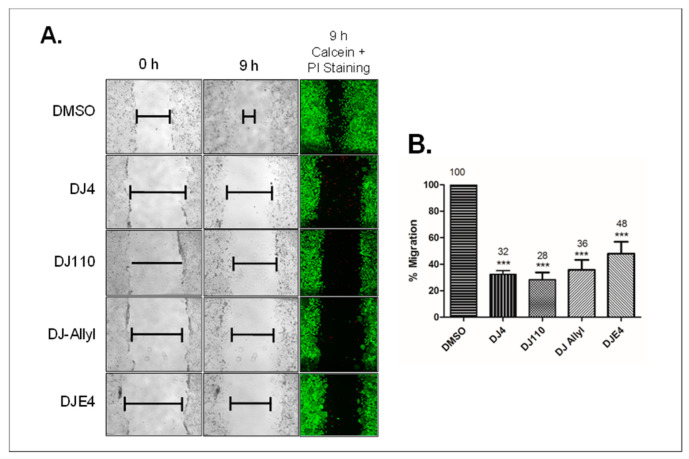
**DJ4 analogs inhibit migration of MDA-MB-231 breast cancer cells.** (**A**) MDA-MB-231 cells were treated with 2.5 µM concentration of DJ4 analogs for 24 h. At the end of the treatment period, uniform scratches were made across the monolayer of cells. The cells were allowed to migrate in the scratched area for 9 h in a compound-free medium. The cells were stained with calcein (green) and propidium iodide (red) to study live and dead cell populations in the scratched area. (**B**) The distance migrated by the treated cells was normalized to vehicle control treated cells. Statistical significance was determined by one-way ANOVA followed by Dunnett’s multiple comparison test. The error bars indicate SEM from triplicates of the experiment. *** *p* < 0.001.

**Figure 10 pharmaceuticals-16-01060-f010:**
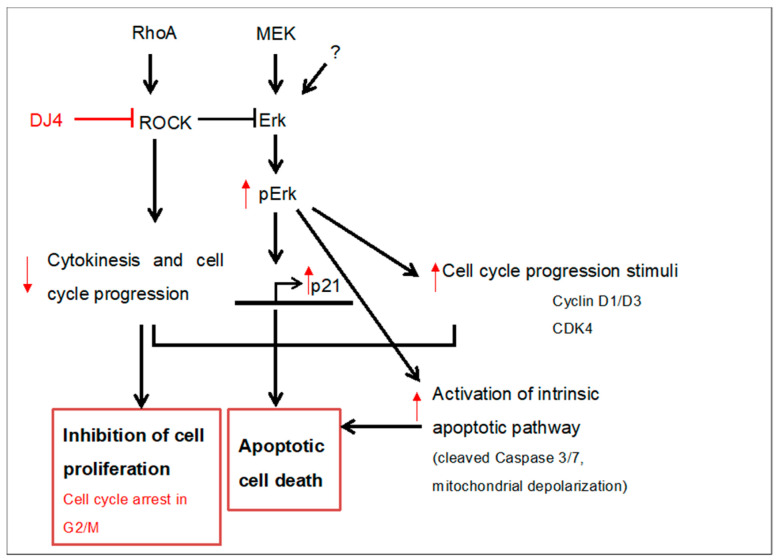
**A proposed model for a probable interplay of DJ4-mediated modulation of pro-proliferative and pro-apoptotic proteins**. DJ4-mediated inhibition of ROCK leads to increased phosphorylation of Erk. Hyper-phosphorylated Erk upregulates p21^WAF1/CIP1^, caspase 3/7, CDK4, and cyclin D3. Additionally, apoptosis can be a result of G2/M arrest, and activation of pro-proliferative proteins and CDK inhibitor p21^WAF1/CIP1^.

**Table 1 pharmaceuticals-16-01060-t001:** Mean cytotoxic concentration of DJ4 in NSCLC cells measured by MTT assay.

Cell Lines	IC_50_ (µM)
H1299	0.44 ± 0.04
H226	0.67 ± 0.38
A549	0.98 ± 0.21
H522	1.69 ± 0.48
H23	2.91 ± 0.71
H460	9.53 ± ND

**Table 2 pharmaceuticals-16-01060-t002:** IC_50_ (μM) values of DJ4 analogs from MTT assays.

Compound	MDA-MB-231	A549	A375M
DJ110	0.65 ± 0.07	0.88 ± 0.16	0.49 ± 0.03
DJ4	2.3 ± 0.23	5.4 ± 1.19	1.1 ± 0.15
DJ-Allyl	4.1 ± 1.23	5.0 ± 2.20	4.7 ± 0.52
DJE4	5.9 ± 1.63	6.1 ± 1.69	5.1 ± 0.32
DJ-Morpholine	>10 ± NA	>10 ± NA	>10 ± NA

(IC_50_ is derived from triplicates. Values are ±SE).

**Table 3 pharmaceuticals-16-01060-t003:** Heat map of cell growth after treatment with DJ4 analogs in NCI-60 human cancer cell line screening.

Panel Name	Cell Line	DJ4	DJ4-HCl	DJ-Allyl	DJE4
**Leukemia**	CCRF-CEM	9.9	5.5	7.1	3.0
HL-60(TB)	6.8	−6.6	−14.5	−22.4
K-562	29.9	28.5	42.1	26.3
MOLT-4	11.4	8.6	5.8	−6.0
RPMI-8226	39.2	38.7	42.7	29.1
SR	5.6	−1.5	−9.5	0.4
**Non-Small Cell Lung Cancer**	A549/ATCC	19.1	8.1	13.2	11.1
EKVX	46.0	40.9	30.5	14.1
HOP-62	19.9	−3.0	−10.8	−26.5
HOP-92	−9.7	11.8	−1.9	4.4
NCI-H226	3.3	22.4	10.7	10.8
NCI-H23	9.1	3.0	6.8	−7.9
NCI-H322M	5.2	19.6	10.2	5.8
NCI-H460	14.1	15.1	2.2	2.4
NCI-H522	26.6	4.3	−5.3	−28.6
**Colon Cancer**	COLO 205	4.8	−6.8	−20.1	−33.3
HCC-2998	53.9	66.2	44.6	10.2
HCT-116	9.7	5.8	6.3	4.0
HCT-15	36.5	33.5	24.7	16.0
HT29	30.6	27.8	26.4	4.3
KM12	13.1	9.3	7.9	6.2
SW-620	9.9	7.8	7.4	6.5
**CNS Cancer**	SF-268	10.4	8.9	−3.3	−5.8
SF-295	22.6	18.7	13.0	5.0
SF-539	0.4	−37.2	−66.2	−64.2
SNB-75	−10.9	−13.3	−67.3	−71.8
U251	15.8	7.6	0.8	−9.5
**Melanoma**	LOX IMVI	16.3	8.3	0.6	1.8
MALME-3M	30.6	44.8	40.2	−6.2
M14	29.9	31.7	24.5	0.7
MDA-MB-435	21.0	18.5	21.1	−0.8
SK-MEL-2	28.8	−4.6	−8.4	−48.7
SK-MEL-28	16.0	0.8	−2.4	−17.2
SK-MEL-5	14.5	−0.2	2.0	9.2
UACC-257	14.6	−1.7	9.6	−15.9
UACC-62	−1.4	−1.3	9.8	−16.0
**Ovarian Cancer**	IGROV1	14.0	25.6	17.0	21.7
OVCAR-3	5.6	3.4	−7.0	−2.1
OVCAR-4	23.8	32.3	34.7	19.5
OVCAR-5	37.8	18.1	5.3	8.7
OVCAR-8	14.0	9.7	10.7	6.3
NCI/ADR-RES	72.9	58.4	42.1	29.7
SK-OV-3	27.0	2.1	−7.0	−10.3
**Renal Cancer**	786-0	9.0	19.7	4.4	3.0
A498	−7.6	−21.3	−34.5	−54.2
ACHN	16.0	−12.0	−1.2	0.7
CAKI-1	37.4	54.8	30.2	20.6
RXF 393	−20.2	−30.1	−41.3	−39.3
SN12C	15.9	33.5	24.0	12.6
TK-10	34.2	18.8	1.7	−3.8
UO-31	25.2	36.6	3.1	7.5
**Prostate Cancer**	PC-3	25.9	19.2	12.6	2.3
DU-145	20.7	15.3	12.7	9.9
**Breast Cancer**	MCF7	4.0	2.0	3.5	4.0
MDA-MB-231	25.5	25.9	19.8	10.7
HS 578T	8.8	8.8	−15.0	−15.1
BT-549	−7.9	5.3	−29.5	−50.4
T-47D	2.6	−1.3	−8.7	−9.7
MDA-MB-468	−14.5	−2.1	1.0	−34.7

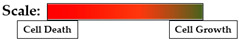
 Positive values indicate percent cell growth and value zero (0) indicate no growth compared to concurrent DMSO treated control at 48 h, while negative values indicate cell death compared to time zero; Cell growth and cell line sensitivity scale.

## Data Availability

Data is contained within the article and Appendix A.

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
