# Peer review of "Characterization of Anticancer Effects of the Analogs of DJ4, a Novel Selective Inhibitor of ROCK and MRCK Kinases"

_pharmaceuticals, 2023, doi:10.3390/ph16081060_

Round 1

Reviewer 1 Report

Non-small cell lung cancer and triple-negative breast cancer cells were used in this investigation to investigate the anti-proliferative and apoptotic effects of DJ4. The authors show that DJ4 causes a cell cycle arrest in the G2/M phase of cancer cells, which triggers the intrinsic apoptotic pathway. Cyclin D3, cyclin dependent kinase 4, and p21WAF1/CIP1 are all proteins involved in cell cycle regulation that DJ4 was shown to upregulate. Authors created a library of 27 analogs to fine-tune DJ4's ROCK/MRCK inhibitory efficacy. They found four functional analogues among the many different structural variants. One drug, DJ110, was found to have increased ROCK/MRCK inhibitory efficacy in cell free kinase activity experiments. In addition, three compounds were shown to be more selective for ROCK1/2 than MRCK/. Cancer cell lines of various types, including non-small cell lung cancer, breast cancer, and melanoma, were used to test the active analogs' ability to inhibit cell proliferation and the cell cycle. Using the NCI-60 human cancer cell line panel, the anti-proliferative efficacy of DJ4 and the active analogs was further proven against a wide variety of cancer cell types. The anti-migratory properties of these novel analogues were finally evaluated in the highly invasive MDA-MB-231 breast cancer cells. The migration of breast cancer cells was decreased by three of the four analogs, with the most effective being DJ4 and DJ110. Overall, the data from this investigation show that selective inhibitors of ROCK1/2 (DJE4, DJAllyl) were less efficient in inducing death than combination inhibitors of ROCK1/2 and MRCK/ (DJ110), despite their ability to suppress cell proliferation and induce cell cycle arrest at G2/M.

There should be clear scheme in 2.1. Scheme for optimization of DJ4. Please start from new figure 1 here and remove the citation of Fig 5 in this section.

Include the control in Table 1 and 2.

 However, DJ-Morpholine did not induce cell cycle arrest possi- 440 bly due to poor solubility. how its proved?

Author Response

We thank the reviewer # 1 for your valauble suggestion and comments.  Our response to the comments are as below: 

Comment 1: There should be clear scheme in 2.1. Scheme for optimization of DJ4. Please start from new figure 1 here and remove the citation of Fig 5 in this section.

Response 1: Authors like to thank the reviewer #1.  As suggested we have edited the figure 5 now and in the revised manuscript scheme is included as Fig 1 under materials and methods section.

Comment 2: Include the control in Table 1 and 2.

Response 2: We thank the reviewer for suggestion. Since the IC50 was calculated with respect to vehicle controls in the respective assay, IC50 value for control is not applicable here. 

Comment 3: However, DJ-Morpholine did not induce cell cycle arrest possibly due to poor solubility. how its proved?

Response 3: DJ4 analogs are insoluble in aqueous solution and we cannot measure the Ksp.  The solubility of DJ4 analogs in DMSO were observed as follows:

DJ4 = 7.0 mg / 0.1mL DMSO

DJ110 = 1.8 mg / 0.1mL DMSO

DJE4 = 2.9 mg / 0.1mL DMSO

DJ allyl = 2.5 mg / 0.1mL DMSO

DJ-Morpholine = 1.7 mg / 0.1 mL DMSO

DJ-Morpholine compound is soluble in DMSO (17 mg / mL) and the concentration used for the experiment was in the solubility range in DMSO, and therefore our observation of cell cycle arrest was not due to poor solubility.

Reviewer 2 Report

This manuscript reports the characterization of cellular the mechanism of action of DJ4, a small molecule ROCK/MRCK inhibitor previously discovered by the Authors, in non-small cell lung cancer (NSCLC) and in triple negative breast cancer (TNBC) cell lines. With the aim of increasing efficacy and potency of the parent compound, a series of DJ4 analogs were generated and four particularly active derivatives were further studied in terms of biochemical and cellular activity: DJ-Allyl, DJE4, DJ110 and DJ-Morpholine. DJ-Allyl resulted to have higher potency than DJ4 towards ROCK1/2, and less active towards MRCKalpha. DJE4 and DJ-Morpholine resulted relatively more selective towards ROCK1/2 only, and DJ-Morpholine resulted less active in all cellular assays probably due to poor solubility. DJ110 resulted to be more potent but less soluble than DJ4.

The results relating to DJ4 analogues, therefore, do not appear encouraging: it is demonstrated that the modifications introduced in the so-called "Group 2" of DJ4 can alter the potency and selectivity between ROCK and MRCK kinase, but at the price of a worsening of the chemical-physical characteristics of these compounds.

The article is quite well written and there are no particular criticisms regarding the experimental setup, but the following points should be addressed in order to render the manuscript acceptable for publication.

1. DJ4 is presented as a dual ROCK/MRCK inhibitor, but the KINOMEscan profile (fig. 4) shows about 80 kinases inhibited above 85% at 10 uM. From this single-dose data it is difficult to extrapolate the real selectivity of DJ4, but it is presumably a promiscuous compound and, as stated by the Authors, susceptible to off-target inhibition of other cellular kinases. The cell cycle blocking effects, in particular, may be related to the most inhibited "off-target" kinases, particularly the CDKs. From the point of view of the cellular mechanism of action, this compound could thus represent a case of "polypharmacology" with the simultaneous engagement of multiple kinases in the cell and explain, for example, why non-linear effects in the dose-modulation are observed in pERK and p21 western blots. The Authors should comment on this possibility for DJ4 and its analogs.

2. For a better understanding of the phenotype it is necessary to analyze some biomarkers of CDK activity, for example phospho-RNApol II (Ser5) and phospho-RNApol II (Ser2), substrates of CDK7 and CDK9.

3. How were the antiproliferative activity data shown in Figure 6 interpolated? Was a Hill equation used? How were the points at the highest concentrations of DJ110 handled? From these data, IC50 values were calculated (in Tab. 2) which show a greater antiproliferative power of DJ110 compared to the other compounds, therefore it is important to see the interpolating curves in these graphs.

4. It is clear that this chemical series might have solubility problems, and some observed "paradoxical" cellular effects are attributed to it (e.g. in the MTT assay curves of DJ110), but there is no formal proof of their poor solubility: are precipitates observed in solution and at what doses? The Ksp of these compounds should be calculated and provided.

5. Did the analysis of DJ4, DJ-Allyl and DJE4 in the NCI60 cell line panel show any correlation between the antiproliferative activity at 10 uM and the doubling time of the cell lines?

Minor points and typos:

1. The article is quite exhaustive of information, but some details are missing regarding the materials used. In particular, it is recommended to add the catalog number of the antibodies used. 

2. Page 5, line 193: please specify the meaning of the acronym "IVCLSP"

3. Page 17, line 467: “At 10 uM, since both DJ110 and DJ-Morpholine were …” should instead read “At 10 uM, both DJ110 and DJ-Morpholine were …”

The quality of the English language is acceptable.

Author Response

We thank the reviewer # 2 for the valauable suggestions and critique.  Please see our response as below:

Comment 1: DJ4 is presented as a dual ROCK/MRCK inhibitor, but the KINOMEscan profile (fig. 4) shows about 80 kinases inhibited above 85% at 10 uM. From this single-dose data it is difficult to extrapolate the real selectivity of DJ4, but it is presumably a promiscuous compound and, as stated by the Authors, susceptible to off-target inhibition of other cellular kinases. The cell cycle blocking effects, in particular, may be related to the most inhibited "off-target" kinases, particularly the CDKs. From the point of view of the cellular mechanism of action, this compound could thus represent a case of "polypharmacology" with the simultaneous engagement of multiple kinases in the cell and explain, for example, why non-linear effects in the dose-modulation are observed in pERK and p21 western blots. The Authors should comment on this possibility for DJ4 and its analogs.

Response 1: We like to thank the reviewer #2 for the valauble comment.  We have added a paragraph with repsonse this in the discussion (page 30 and 31). “Moreover, since DJ4 analogs are selective multi-kinase inhibitors, the contribution of inhibition of other kinases not studied, herein, cannot be negated.” It’s hard to say that non-linear effects observed in pERK and p21 are due to multikinase effect. However, we think, as mentioned in the discussion section, that decreased pERK and p21 as higher concentration may be due to higher % of dead cell at 10 uM.

Comment 2. For a better understanding of the phenotype it is necessary to analyze some biomarkers of CDK activity, for example phospho-RNApol II (Ser5) and phospho-RNApol II (Ser2), substrates of CDK7 and CDK9.

Response 2:  Thank you reviewer # 2 for the excellent suggestion. We agree that it is important to analyze inhibitory activity of DJ4 analogs toward other kinases including CDK7/CDK9. However, as we mentioned in the text (Section 3.6), DJ4/analogs are ATP competitive inhibitors, and therefore, it is expected that multiple kinases would be affected as indicated by the KINOMEscan profiling assay data. We will examine those targets in our continuing future studies. With this in mind and as the current manuscript focus on ROCK/MRCK kinases as a follow-up study of our original published study (Ref 11), we believe examining the activity towards CDKs are beyond the scope of this manuscript.

Comment 3. How were the antiproliferative activity data shown in Figure 6 interpolated? Was a Hill equation used? How were the points at the highest concentrations of DJ110 handled? From these data, IC50 values were calculated (in Tab. 2) which show a greater antiproliferative power of DJ110 compared to the other compounds, therefore it is important to see the interpolating curves in these graphs.

Response 3: As per the reviewer suggestion, Figure 6 was revised with Dose-response curves. IC50 values were calculated with nonlinear least squares regression curve fit by GraphPad prism (as indicated in the figure legend and methods). The highest concentrations of DJ110 were omitted from final IC50 analysis due to precipitation in the well (which increased the absorbance values). Thanks for your valuable comment to address confusion with interpolating curves for IC50 calculation.

Comment 4. It is clear that this chemical series might have solubility problems, and some observed "paradoxical" cellular effects are attributed to it (e.g. in the MTT assay curves of DJ110), but there is no formal proof of their poor solubility: are precipitates observed in solution and at what doses? The Ksp of these compounds should be calculated and provided.

Response 4: DJ4 analogs are insoluble in aqueous solution and we cannot measure the Ksp. The solubility of DJ4 analogs in DMSO were observed as follows:

DJ4 = 7.0 mg / 0.1mL

DJ110 = 1.8 mg / 0.1mL

DJE4 = 2.9 mg / 0.1mL

DJ allyl = 2.5 mg / 0.1mL

DJ-Morpholine = 1.7 mg / 0.1 mL

Comment 5. Did the analysis of DJ4, DJ-Allyl and DJE4 in the NCI60 cell line panel show any correlation between the antiproliferative activity at 10 uM and the doubling time of the cell lines?

Response 5: We reviewed the doubling time of these cell lines (https://dtp.cancer.gov/discovery_development/nci-60/cell_list.htm) and anti-proliferative activity of the analogs in these cell lines. However, we do not think there is any direct correlation of doubling time and the anti-proliferative activity observed in these cell lines. 

Minor points and typos:

  1. The article is quite exhaustive of information, but some details are missing regarding the materials used. In particular, it is recommended to add the catalog number of the antibodies used. 

Response: We thank the reviewer for the suggestions and in the revised manuscript the catalogue numbers were recorded for the antibodies.

  1. Page 5, line 193: please specify the meaning of the acronym "IVCLSP"

 Response: We thank reviewer and in the revised manuscript, the acronym was explained.

  1. Page 17, line 467: “At 10 uM, since both DJ110 and DJ-Morpholine were …” should instead read “At 10 uM, both DJ110 and DJ-Morpholine were …”

Response: Authors thank the reviewer for the suggestions accordingly the sentence was corrected in the revised manuscript.

Round 2

Reviewer 2 Report

The manuscript may be considered for publication in the present revised form.

The quality of English language is adequate.